# NK Cells Acquire CCR5 and CXCR4 by Trogocytosis in People Living with HIV-1

**DOI:** 10.3390/vaccines10050688

**Published:** 2022-04-28

**Authors:** Dang-Nghiem Vo, Nicolas Leventoux, Mauricio Campos-Mora, Sandrine Gimenez, Pierre Corbeau, Martin Villalba

**Affiliations:** 1IRMB, University Montpellier, INSERM, 34295 Montpellier, France; dang_nghiem.vo@med.lu.se (D.-N.V.); mauricio-alejandro.campos-mora@inserm.fr (M.C.-M.); 2Institute for Human Genetics, CNRS-Montpellier University, UMR9002, 141 Rue de la Cardonille, CEDEX, 34396 Montpellier, France; nicolas.leventoux@igh.cnrs.fr (N.L.); sandrine.gimenez@igh.cnrs.fr (S.G.); 3Immunology Department, University Hospital, Place du Pr Debré, CEDEX, 30029 Nîmes, France; 4IRMB, University Montpellier, INSERM, CNRS, CHU Montpellier, 34295 Montpellier, France; 5Institut du Cancer Avignon-Provence Sainte-Catherine, 84000 Avignon, France

**Keywords:** NK cells, HIV-1, trogocytosis, CCR5, CXCR4

## Abstract

NK cells play a major role in the antiviral immune response, including against HIV-1. HIV-1 patients have impaired NK cell activity with a decrease in CD56^dim^ NK cells and an increase in the CD56^−^CD16^+^ subset, and recently it has been proposed that a population of CD56^+^NKG2C^+^KIR^+^CD57^+^ cells represents antiviral memory NK cells. Antiretroviral therapy (ART) partly restores the functional activity of this lymphocyte lineage. NK cells when interacting with their targets can gain antigens from them by the process of trogocytosis. Here we show that NK cells can obtain CCR5 and CXCR4, but barely CD4, from T cell lines by trogocytosis in vitro. By UMAP (Uniform Manifold Approximation and Projection), we show that aviremic HIV-1 patients have unique NK cell clusters that include cells expressing CCR5, NKG2C and KIRs, but lack CD57 expression. Viremic patients have a larger proportion of CXCR4^+^ and CCR5^+^ NK cells than healthy donors (HD) and this is largely increased in CD107^+^ cells, suggesting a link between degranulation and trogocytosis. In agreement, UMAP identified a specific NK cell cluster in viremic HIV-1 patients, which contains most of the CD107a^+^, CCR5^+^ and CXCR4^+^ cells. However, this cluster lacks NKG2C expression. Therefore, NK cells can gain CCR5 and CXCR4 by trogocytosis, which depends on degranulation.

## 1. Introduction

The primary role of Natural Killer (NK) cells in the host innate immune system is to provide first line of protection against viral infection and other malignancies via the mechanism of immune-surveillance [1]. During steady states as well as in acute viral infections, NK cells are thought to constantly interact with their neighboring cells within the tissue microenvironment or in circulation via a wide range of receptors and adhesion molecules expressed on their surface [1,2,3]. As a consequence, these close-range contacts between NK cells and their targets often results in passive or active transferring of membrane patches and other surface molecules from the target cells into the effector (NK) cells [4], a process called trogocytosis [5,6]. Trogocytosis is a process firstly described and widely observed in other lymphocytes including B and T cells [5]. The potential trogocytosis between HIV-1-infected cells and NK cells, as well as its physiological consequence, has not been described, and very little is known regarding other immune cells [6,7].

We have identified a population of endogenous NK cells that is actively fighting against tumor cells in cancer patients [4,8,9,10]. In hematological cancers, i.e., multiple myeloma, acute myeloid leukemia, B cell lymphoma and B-chronic lymphocytic leukemia, this CD56^dim^CD16^+^ population is highly activated and has recently degranulated, which is evidence of killing activity. These NK cells, which have been engaged in tumor cell killing, can be easily distinguished by their CD45RA^+^RO^+^ phenotype (CD45RARO cells), as opposed to non-activated cells in patients or in healthy donors displaying a CD45RA^+^RO^−^ (CD45RA cells) phenotype similar to naïve T cells [4]. A high percentage of CD45RARO cells expressed NK cell p46-related protein (NKp46), natural-killer group 2, member D (NKG2D) and killer inhibitory receptors (KIRs) and a low percentage expressed NKG2A and CD94. They are also characterized by a high metabolic activity and active proliferation [4]. We observed that some patients’ NK cells have performed an astonishingly efficient trogocytosis and capture of antigens from target cells, i.e., CD19 from the tumor B cells of a B-cell lymphoma patient or CD14 from the tumor cells of an AML patient and so on [4,9,10]. NK cells basically uptake antigens from tumor cells in all studied hematological malignancies.

Patients with impaired NK cell functions are susceptible to viral infections [11]. Particularly, NK cells are able to kill HIV-1-infected CD4+ T cells [12,13], and they play a role in HIV-1 containment [14]. The aim of this work was to test whether, by killing CD4^+^CCR5^+^ and/or CXCR4^+^ infected T cells, NK cells might acquire HIV-1 receptor and/or coreceptor(s) via trogocytosis.

## 2. Materials and Methods

### 2.1. Patient Samples

People living with HIV-1 and between 55 and 70 years of age were proposed to participate during a routine visit at the University Hospital of Nîmes, France. Patients were either nontreated and viremic or under stable antiretroviral therapy with a viral load below 50 copies per mL for at least six months. Their CD4 counts were ≥200 cells/µL. Pregnant or breastfeeding women, persons presenting a treatment or another disease modifying their immune system were not included. This study was approved by a National Ethics Committee with the identification code: Ref 13 06 03 N° ID-RCB: 2013-A00795-40. All patients provided written informed consent. The trial was registered on ClinicalTrials.gov (accessed on 10 March 2022) (NCT02592174) (https://clinicaltrials.gov/ct2/history/NCT02592174?V_4=View).

### 2.2. Cell Lines

The MT4 are derived from Adult T-cell Leukemia patients and carry HTLV-1. The CEM is derived from an acute lymphoblastic leukemia (ALL) patient. Both cell lines were grown in RPMI 1640 supplemented with 10% fetal bovine serum, 2 mM glutamine and penicillin/streptomycin.

### 2.3. Cell Preparation

Expansion of primary NK cells were obtained by stimulating 1 × 10^6^ PBMCs/mL during 14 to 20 days with the lymphoblastoid EBV-positive cell line PLH together with IL-2 (100 U/mL, PrePotech) and IL-15 (5 ng/mL, Miltenyi), as previously described [15]. Primary CD56-positive cells were purified from healthy donor buffy coat using the kit EasySep Buffy Coat CD56 Positive (Stemcell) according to the manufacturer’s instructions.

### 2.4. Coculture

For in vitro trogocytosis assay, the T cell lines CEM and MT4 were labelled with CelltraceViolet (Thermo Fisher, Waltham, MA, USA) following the manufacturer’s protocol to allow identification as a target cell. CTV-labelled target cells were next incubated with primary NK cells derived from PBMC. Upon 4 h in vitro incubation, CCR5 and CD4 expressions on NK cells were analyzed by FACS analysis. CTV-positivity on NK cells indicated interaction between effector (NK) and target (CEM or MT4) cells.

### 2.5. Flow Cytometry for In Vitro Trogocytosis Analysis

To analyze CXCR4 acquisition by the NK cell line, lymphocytes were stained for one hour at room temperature in the dark with CD56-P5.5 (Beckman Coulter, Brea, CA, USA) and CXCR4-PE (BD Biosciences, Franklin Lakes, NJ, USA). To analyze CXCR4, CD3, and CD4 acquisition by primary NK cells, lymphocytes were stained for one hour at room temperature in the dark with CD56-APC (Miltenyi, Paris, France) and CXCR4-PE (BD Biosciences) or CD3 APC-cy7 (Sony, Minato, Tokyo, Japan) or CD4-FITC or -PE (Beckman Coulter). To analyze CCR5 acquisition by primary NK cells, lymphocytes were stained for one hour at room temperature in the dark with VPD450 (BD Biosciences), CD56-APC (Miltenyi), CD3 APC-cy7 (Sony), and CCR5-PE/Cy5.5 (BD Biosciences). Cells were fixed using Cellfix (BD biosciences).

### 2.6. FACS Analysis and Antibodies

For immune-monitoring studies, patient PBMC samples were stained with cocktail fluorescent-conjugated antibody mix diluted in PBS containing 2% FBS + 0.01% azide for 30 min at 4 °C in the dark. After a 2-step washing with PBS, cells were fixed with 2% PFA for 20 min at 4 °C. Samples were analyzed on a BD Fortessa Flow Cytometer (UV-Violet-Blue-Yellow/Green-Red 5-Laser configuration). For the first cohort (aviremic patients) the following antibodies were used: CCR5-FITC, CD158a-PE, CD57-PE-CF594, CD14-Alexa Fluor 700, CD19-Alexa Fluor 700, CXCR4-BV421, CD62L-BV605, NKG2D-BV650, CD16-BV711, CD3-BV786, CD56-BUV395 and CD4-BUV737 (BD Biosciences), CD158b-PE, CD158e/k-PE, NKp46-PE-Vio770, NKG2C-APC, CD45RA-APC-Vio770, and CD45RO-VioGreen (Miltenyi). For the second cohort (viremic patients) the similar panel of antibodies were re-used, except for additional antibodies for CXCR4-BV421 and CD107a-PE-CF594 (BD Biosciences). Cell death and viability were determined by DAPI (BD Biosciences). Data were analyzed using FlowJo software (v10.8) (BD Biosciences) with appropriate plugins for high-dimensional analysis and visualization.

### 2.7. High Dimensional Reduction Analysis

High-dimensional reduction analyses (t-SNE and UMAP) were performed as previously described [9]. Briefly, total NK cells for each sample were selected by manual gating to exclude dead/debris cells, and other lineages (myeloid/B cells) and gated on CD3^−^CD56^+^ population. The number of NK cells for each sample was established at 40,000 using DownSample (v3.1) and eventually merged into one unique FCS file with the concatenation function. Finally, the dimensionality reduction algorithms UMAP (UMAP v2.2) was performed and the specific maps derived from healthy donors or patients were generated based on SampleID parameters created during the concatenation step.

### 2.8. Cell Transduction

We have previously described the HIV-1 vectors delivering the *CCR5/EGFP* and the *LacZ* genes and the production of the corresponding transducing particles [16].

### 2.9. Statistical Analysis

Figures presented in the manuscript and appropriate statistical analysis were performed using GraphPad Prism (v9.2.0). A detailed description of the respective statistically significant test is indicated directly under each figure legend, with appropriate post-hoc correction tests applied. For all figures, statistical significances were presented as * *p* < 0.05; ** *p* < 0.01; *** *p* < 0.001 and **** *p* < 0.0001. Mean values are expressed as mean plus or minus the standard error of the mean (SEM).

## 3. Results

### 3.1. Human NK Cells Cocultured with T Cell Lines Acquire T Cell Markers

We have previously shown that human NK cells perform efficient trogocytosis in several immune cell types, e.g., myeloid or B cells [4,8,9,10]. To examine if NK cells can perform trogocytosis on T cells, we isolated NK cells from a healthy donor and co-cultured them with the T cell lines CEM and MT4 that were previously labeled with the cell tracker violet (CTV; Appendix A). CEM and MT4 both expressed CD4 (partially for CEM). MT4 express the chemokine receptor CCR5. After 4 h of co-culturing, a proportion of NK cells were CTV^+^ (Appendix A). In addition, some CTV^+^ NK cells also gained expression of CD4 after incubation with both cell lines (Appendix A). CCR5 was expressed by NK cells only after incubation with MT4 cells (Appendix A).

### 3.2. NK Cells Cocultured with CXCR4^+^ Target Cells Acquire CXCR4 Surface Expression

Next, we tested whether human NK cells were able to acquire the chemokine receptor CXCR4 upon contact with virus-infected CXCR4-positive cells. To this aim, we prepared expanded NK cells (eNK) by culturing PBMCs with the lymphoblastoid EBV-positive PLH cell line in the presence of IL-2 and IL-15 [15,17,18]. Then, we cocultured these eNK with the MT4 cell line. Almost no CXCR4 molecule was detected at the surface of NK cells cultured alone (Appendix A). CXCR4 expression increased on NK cells cocultured for 18 h with MT4 cells (Appendix A).

We performed the same experiment with naïve CD56-positive lymphoid cells directly purified from PBMCs. Again, CXCR4 appeared at the surface of these cells after an overnight coculture with MT4 cells, but not at the surface of monocultured cells (Figure 1A,B). Of note, 62% of the cocultured CD56-positive lymphoid cells became CXCR4-positive. Expression of the CD4 molecule on NK cells was much lower (Figure 1C,D).

### 3.3. Trogocytosis Is Responsible for CXCR4 Acquisition by Cocultured NK Cells

Next, we wanted to ascertain that the appearance of this chemokine receptor on NK cells cultured in the presence of CXCR4-positive cells was due to a cell-to-cell transfer of CXCR4, rather than, for instance, the expression of the *CXCR4* gene in activated NK cells or to the translocation of CXCR4 from the inner of the NK cells to their surface. In addition, we wished to investigate if NK cells can obtain other HIV-1 coreceptors from T cells. For this purpose, we transduced MT4 cells with HIV-1 vectors delivering the *CCR5* gene fused to the marker gene *EGFP* to obtain CCR5/EGFP-MT4 cells expressing CCR5/EGFP fusion protein (Figure 2A). As a negative control, we transduced in parallel MT4 cells with an HIV-1 vector harboring the *LacZ* gene (LacZ-MT4 cells). We labeled freshly purified PBMC cells with the vital dye VPD to rigorously distinguish NK cells from MT4 cells. Then, we cultured these PBMC either with CCR5/EGFP-MT4 or with LacZ-MT4 cells. Eighteen hours later, we analyzed EGFP and surface CCR5 expression on/in VPD^+^CD56^+^CD3^−^ cells. Figure 2B,D shows that 25% of NK cells became EGFP^+^ in the presence of CCR5/EGFP-MT4 cells but not in presence of LacZ-MT4 cells. These data show that NK cells acquire cell surface expression of the chemokine receptor CCR5, and that this acquisition is the consequence of a transfer from the target to the NK cell.

To ascertain that the transfer of CCR5/EGFP from the MT4 cells to the NK cells was direct, we repeated the experiment with freshly sorted primary NK cells instead of PBMC. Here again, we observed the appearance of CCR5/EGFP at the surface of the purified NK cells cocultured with CCR5/EGFP-MT4 cells, but not at the surface of NK cells cocultured with LacZ-MT4 cells (Figure 2E–G).

### 3.4. UMAP (Uniform Manifold Approximation and Projection) Identifies NK Cell Subsets That Have Actively Performed Trogocytosis and Degranulated in HIV-1 Aviremic Patients under Treatment

To more precisely characterize the NK cell population in healthy donors (HDs) and patients, we used the high dimensional reduction algorithm UMAP to generate unsupervised UMAP embed maps as previously described [9,19]. We used 12 NK surface markers (Appendix A) to generate a UMAP plot from 15 samples, i.e., 11 aviremic HIV-1^+^ patients and four HDs. Each sample contained 40,000 NK cells (Figure 3A). From this original map, we derived two daughter maps only showing the events from HD or patients (Figure 3B), and we identified four clusters. Two of them were common and represented CD56^dim^ and CD56^bright^ NK cell populations. The two other clusters were specific to patients and could represent more than 30% of the total NK cells (Figure 3C).

Both of these clusters expressed high levels of CD16 and KIRs and also NKG2C (Figure 4A). However, only HIV-1 cluster 1 expresses CD57 (Figure 4A,B). In view of the relationship between a viral infection, i.e., CMV, and NKG2C expression [20,21], we directly analyzed NKG2C expression in our samples. Of note, although HIV-1 patients harbor CMV more often than HIV-1-negative persons, CMV does not induce CXCR4 expression [22]. Compared with our HD cohort, all patients had higher NKG2C^+^ NK cell populations (Figure 4C). CCR5 expression on NK cells from patients was higher than that from HD (Figure 4C). HIV-1 cluster 2 expressed the highest percentage of CCR5^+^ cells, and this cluster did not express CD57. NK cells lacked CD4 expression independently of any cluster (Figure 4A).

### 3.5. Increased Expression of CCR5 and CXCR4 by NK Cells in Viremic HIV-1 Patients

We hypothesized that NK cells in contact with T cells freshly infected with HIV-1 would have increased trogocytosis, thus we investigated if HIV-1 patients with current viremia had NK cells expressing cell surface CCR5 and CXCR4. We analyzed blood samples from a second cohort of four healthy donors (HD) and 4 HIV-1 patients (Figure 5). Almost 40% of the CD56^+^CD3^−^ NK cells derived from patients expressed CXCR4 and more than 20% were also CCR5^+^ (Figure 5B). These values were significantly higher than those found in HD. These NK cells from patients also expressed higher levels of the degranulation marker CD107a.

During chronic HIV-1 infection, there is an expansion of a CD56^−^ NK cell population that can constitute up to half of the peripheral NK cells and are functionally impaired [23,24,25]. This NK cell subset is also found in healthy individuals [26,27], but it is unknown to what extent this population represents a similar or distinct phenotype in patients and in healthy individuals. Hence, we analyzed the expression of the abovementioned markers in the CD56^−^CD16^+^ population. We observed a significant increase in CXCR4 and CCR5 expression (Figure 5), but patients were highly heterogenous regarding this population and the markers expressed.

We next analyzed the correlation between the cells that have degranulated and those that have gained CXCR4 and CCR5. In patients, most of the cells that had degranulated, were CXCR4- and CCR5-positive (Figure 6A–C). In contrast, only around 20% of the cells that had not degranulated expressed these receptors. Figure 6C shows the statistical relevance of these observations. In summary, trogocytosis is linked to degranulation.

We used the same approach in the CD56^−^CD16^+^ population. CXCR4 and CCR5 MFI values were much lower in this population compared with CD56^+^ cells (compared in Figure 6A,D). The CD56^−^CD16^+^ population derived from patients showed higher trogocytosis in the cells that had degranulated, but the differences were lower than for CD56^+^ cells (Figure 6D). In HD the values were lower, mainly in the cells that had not degranulated (Figure 6E). Figure 6F showed the statistical relevance of our results.

### 3.6. UMAP (Uniform Manifold Approximation and Projection) Identifies NK Cell Subsets That Have Actively Performed Trogocytosis and Degranulated

To more precisely characterize the NK cell population in HD and viremic patients, we also used the high dimensional reduction algorithm as described above. This approach clearly distinguished the different immune cell populations and showed a decrease in CD4 T cells and, in less proportion, NK cells in viremic HIV-1 patients (Appendix A).

Next, we used 12 NK surface markers (Appendix A) to generate a UMAP plot for the eight samples, i.e., 4 patients and 4 HDs. Each sample contained 40,000 NK cells (Figure 7A). From this original map, we derived two daughter maps only showing the events from HD or patients. Patients showed UMAP maps that were relatively homogenous and clearly distinct from that of HDs (Figure 7B). The common cluster contained the CD56^bright^ population, which usually expresses low CD16 and Killer Immunoglobin-like Receptors (KIRs) (Figure 7C). The HD clusters contained matured NK cells with HD cluster 2 expressing high CD62L (Figure 7C). HIV-1 clusters 1 and 2 are very similar mainly comprising mature NK cells with relatively high KIR expression. However, cluster 2 contained cells that expressed NKG2C. Cluster 3 was formed by CD56^dim^ cells with few cells expressing CD16 or NKG2C and 50% expressing KIRs. They also expressed also low CD45RA levels. Interestingly, this cluster contained the highest proportion of cells that had degranulated. In addition, they expressed CCR5 and CXCR4.

We next performed a similar UMAP analysis in the CD56^−^CD16^+^ cells (Appendix A) and identified two HDs and two patients’ clusters (Figure 8A). Individual patient’s maps were heterogenous (Figure 8B). CD107a, CCR5 and CXCR4 expressions were higher in patients and did not correlate with NKG2C and KIR expressions (Figure 8C). Hence, in both NK cell populations, degranulation and potential trogocytosis were not linked to NKG2C expression in viremic patients. Therefore, we investigated in the whole NK cell population the correlation between NKG2C and CD107a. Cells lacking NKG2C showed increased degranulation (Appendix A) and CD107a^−^ cells express more NKG2C than CD107a^+^ cells. Therefore, in our study the NK cell population that was degranulating and possibly performing trogocytosis on T cells did not express NKG2C at the time of analysis.

## 4. Discussion

In the present study, we show evidence of the transfer of the chemokine receptors CXCR4 and CCR5 from virus-infected target cells towards NK cells. Of note, CCR5 transfer to PBMC has already been reported, but via microparticles [28]. In our work, trogocytosis appeared to be selective, confirming previous observations [29]. Contrary to an earlier work claiming the capture of CD4 by CD8+ T cells via trogocytosis [30], here we observed that NK cells barely capture CD4 molecules. Why are certain plasma membrane molecules trogocytosed and not others? This is unknown, and it is probably related to the unknowledge of the mechanism(s) of trogocytosis [5,31]. Of note, the co-trogocytosis of CD4 with CCR5 or CXCR4 could have resulted in the HIV-1 infectibility of NK cells in the same way as trans-synaptic acquisition of CD21 by NK cells allowed EBV binding [32]. Moreover, NK cells can eliminate EBV bound to B cells through a specific antibody-mediated uptake [33]. It would be interesting to know if this is the case of HIV-1 from T cells.

It has already been shown that transferred membrane receptors after trogocytosis may be functional and modify the functions of the acquirer cell [34]. For instance, transferred Ig-like Transcript 2 on T cells that acquired them via trogocytosis are able to signal and modify the functions of their new host [35]. The newly acquired chemokine receptors CXCR4 and CCR5 may be functional at the surface of the cytotoxic cells, as previously reported for CCR7 [36]. This could result in a modification of NK cell circulation and homing. Such a modification could impact NK cell functionality, inasmuch as other molecules, as adhesion molecules, for instance, may have also been captured. It could also affect NK cell function in different therapies [37]. This risk has to be taken into consideration and to be further explored for the therapeutic use of chimeric antigen receptors (CAR)-NK cells.

We have observed that the amount of antigen and the percentage of cells that have performed trogocytosis is higher in the CD56^+^ population than in the CD56^−^CD16^+^ subset. Taking the previous literature into account [23,24,25,38], we suggest that the CD56^−^CD16^+^ cells that have trogocytosed or degranulated are exhausted NK cells that have performed their function and would not perform a cytotoxic function any longer and are losing the trogocytosed markers.

We observed an increase in cells expressing CCR5 compared to HD in aviremic patients. This observation supports the hypothesis that NK cells could gain antigens from HIV-1-infected T cells that are still producing viral components [38,39]. The control of HIV-1-1 viremia by ART also leads to the recovery of NK cells with cytotoxicity against HIV-1 infected T cell targets [39,40]. In agreement, the cells present in HIV-1 cluster 2 (described here in Figure 4) exhibit NKG2C and KIRs and have performed trogocytosis on CCR5. This resembles the recently described memory-like NK cell population in the HIV-1 patients or the macaques model [20,21,39,41,42,43], which shows increased effector functions when reencountering viral antigens. This would explain why they carry T cell markers, i.e., they obtained them by trogocytosis after encountering HIV-1-infected T cells. However, we observed the cells that have trogocytosed lacked CD57 expression. It is possible that when evolving from memory (CD57^+^) to effector cells, CD57 is lost. Interestingly, in active viremic patients, we found a negative relationship between degranulation and trogocytosis *versus* NKG2C expression. It is probable that these patients have not yet generated NKG2C^+^ memory-like NK cells or, in active viremia, this population did not have time to establish itself. Interestingly, in viremic patients the cluster that has degranulated and performed trogocytosis are CD56^dim^ cells with low CD16 expression. This decrease in CD16 is found in NK cells after encountering target cells [44]. They also expressed low CD45RA levels, which could be due to the increased expression of CD45RO. CD45RO is linked to antitumor NK cell function in hematological cancer patients [4,8,9,10]. In summary, in HIV-1 patients the populations of memory and effector NK cells are phenotypically different. Antiviral memory NK cells express NKG2C, whereas antiviral effector NK cells lack NKG2C, but can be recognized by degranulation and by their capacity to perform trogocytosis.

Trogocytosis is becoming a field of intense study to understand the communication/interaction of tumor and immune cells which can affect the clinical outcome as recently shown by the trogocytosis of PD-1 by NK cells in cancer patients [45]. Being sure that ex vivo observations properly mirror the in vivo physiology is extremely difficult for all of these types of studies, and hence, our study has some limitations. Besides trogocytosis, CCR5 and CXCR4 overexpression on NK cells, we report here that nontreated HIV patients could have other causes. A genetic cause is unlikely. Δ32CCR5 heterozygosity is a known cause of low cell surface CCR5 density. Yet, the frequency of Δ32CCR5/WTCCR5 individuals in France is 0.18 [46], so the probability for the 4 HD of each cohort that we have analyzed to be Δ32CCR5 heterozygous is extremely low. Coinfections could also play a role. For instance, human CMV infection, which is more frequent in HIV-positive than in HIV-negative individuals, is associated with an increased frequency of NKG2C^+^ NK cells [47]. Yet, as mentioned, CMV does not induce CXCR4 expression. It would be interesting to support our proposal that CCR5 and CXCR4 overexpression on NK cells in HIV patients is due to trogocytosis by quantifying *CCR5* and *CXCR4* mRNA in NK cells to discard the hypothesis of an endogenous production. However, this implies the sorting of a small number of cells and the corresponding ex vivo manipulation that could affect gene expression.

## 5. Conclusions

NK cells play a major role in the antiviral immune response, including against HIV-1. We identified specific NK cell subsets that expressed CCR5 and CXCR4, but barely CD4, T cell antigen markers on their plasma membrane. We show a strong association of degranulation and trogocytosis ex vivo, suggesting that NK cells are eliminating or trying to eliminate HIV-1 infected T cells. By UMAP (Uniform Manifold Approximation and Projection), we show that aviremic HIV-1 patients have unique NK cell clusters that encompass cells expressing CCR5, NKG2C and KIRs, but lack CD57 expression. Viremic patients have a larger proportion of CXCR4^+^ and CCR5^+^ NK cells than healthy donors (HD), and this subset lacks NKG2C expression. Therefore, our results strongly suggest that NK cells can gain CCR5 and CXCR4 by trogocytosis in vivo, which depends on degranulation and does not always correlate with NKG2C expression.

## Figures and Tables

**Figure 1 vaccines-10-00688-f001:**
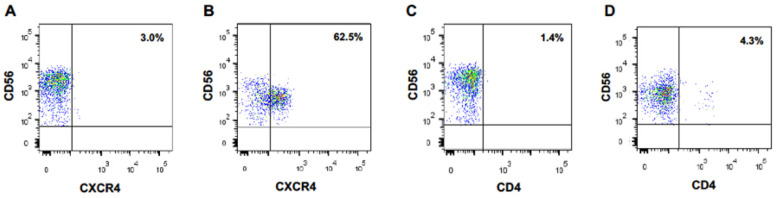
CD56-positive lymphoid cells acquire CXCR4 cell surface expression upon coculture with CXCR4-positive MT4 cells. CXCR4 (**A**) and CD4 (**C**) expression at the surface of primary CD56^+^ lymphocytes cells cultured alone. CXCR4 (**B**) and CD4 (**D**) expression at the surface of primary CD56^+^ lymphocytes cocultured overnight with MT4 cells.

**Figure 2 vaccines-10-00688-f002:**
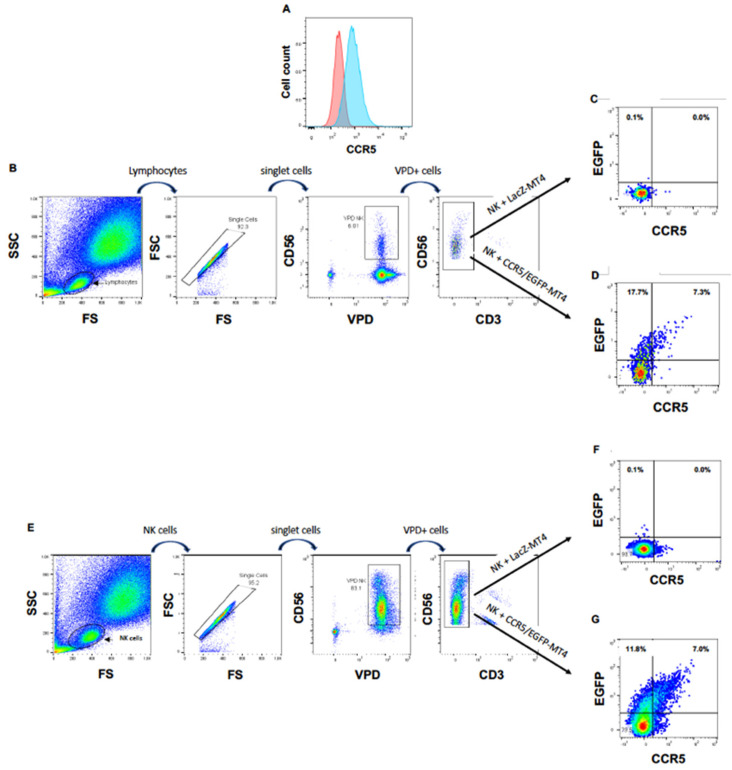
NK cells acquire CCR5 by trogocytosis. (**A**) CCR5 expression on non-transduced (left hand histogram) and CCR5/EGFP-transduced (right hand histogram) MT4 cells. (**B**) Gating strategy for VPD-labeled PBMC cocultured with MT4 cells. (**C**) EGFP and CCR5 expression of NK cells among PBMC cocultured with LacZ-transduced MT4 cells. (**D**) EGFP and CCR5 expression of NK cells among PBMC cocultured with CCR5/EGFP-transduced MT4 cells. (**E**) Gating strategy for VPD-labeled NK cells directly cocultured with MT4 cells. (**F**) EGFP and CCR5 expression of NK cells cocultured with LacZ-transduced MT4 cells. (**G**) EGFP and CCR5 expression of NK cells directly cocultured with CCR5/EGFP-transduced MT4 cells.

**Figure 3 vaccines-10-00688-f003:**
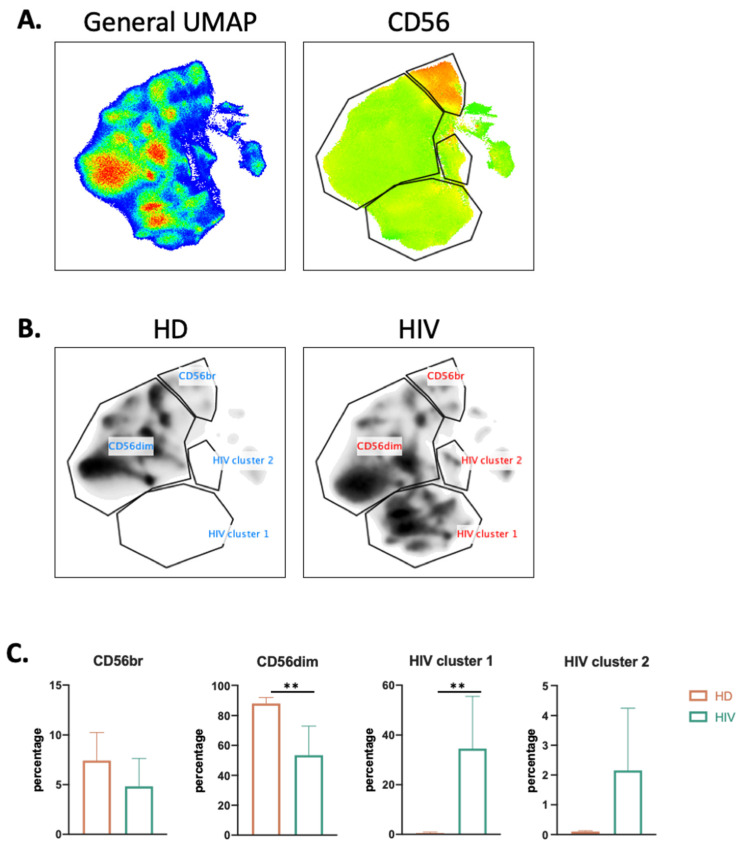
UMAP analysis of second cohort of HIV-1^+^ patients. (**A**) UMAP plot was generated from 40,000 pre-gated NK cells (CD14-CD19-CD3-CD56^+^ phenotype) from a cohort of HIV-1 patients (*n* = 11) and HD (*n* = 4), in the right CD56 heatmap of CD56 expression on different populations. (**B**) UMAP specific of HD and HIV-1+ patients. (**C**) Proportion of each cluster among HD (*n* = 4) and HIV-1 patients (*n* = 11). Statistical significance was determined by unpaired *t*-test; ** *p* ≤ 0.01.

**Figure 4 vaccines-10-00688-f004:**
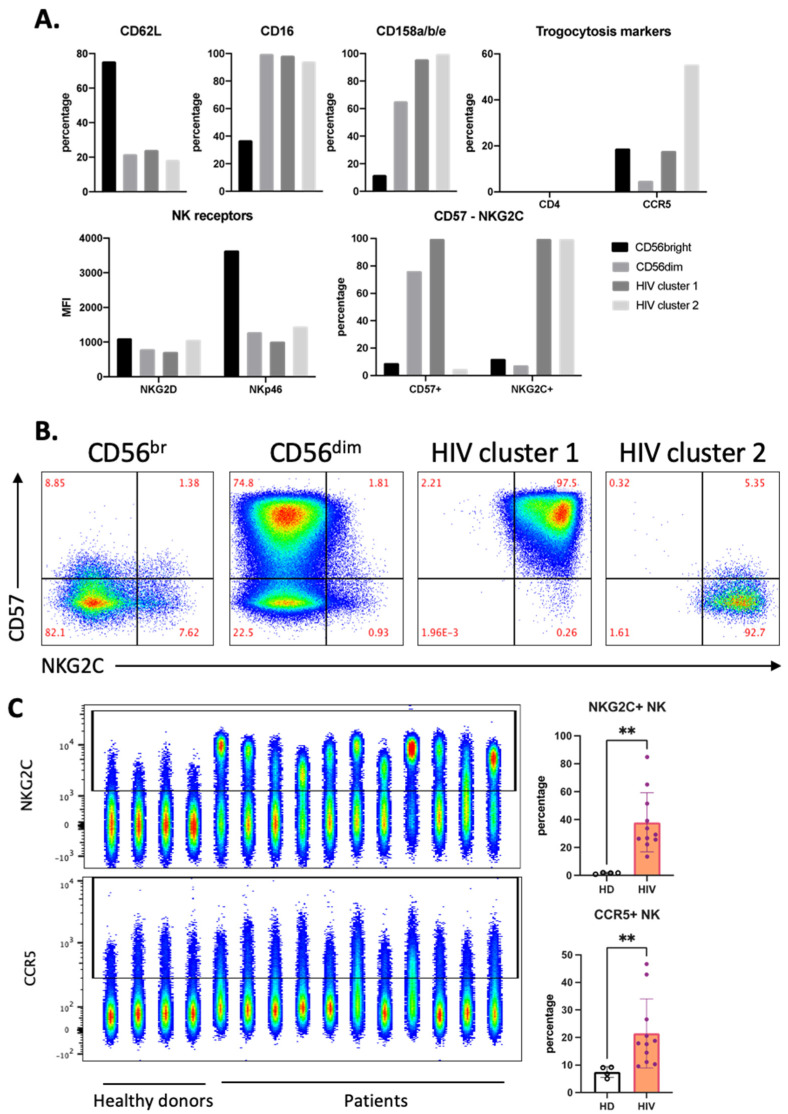
Distinct phenotype of HIV-1-specific clusters in aviremic HIV-1 patients. (**A**) Phenotype of each cluster identified in Figure 3 by percentage and expression level (MFI) of each marker in the panel. (**B**) Manual gating on CD57 and NKG2C for all concatenated NK cells from the HIV-1 patients, which were divided into four clusters, as described in Figure 3B. (**C**) Accumulation of NKG2C^+^ and CCR5^+^ NK in HIV-1 patients represented by FACS plot (left) and summarized statistics (right). Statistical significance was determined by unpaired *t*-test; ** *p* ≤ 0.01.

**Figure 5 vaccines-10-00688-f005:**
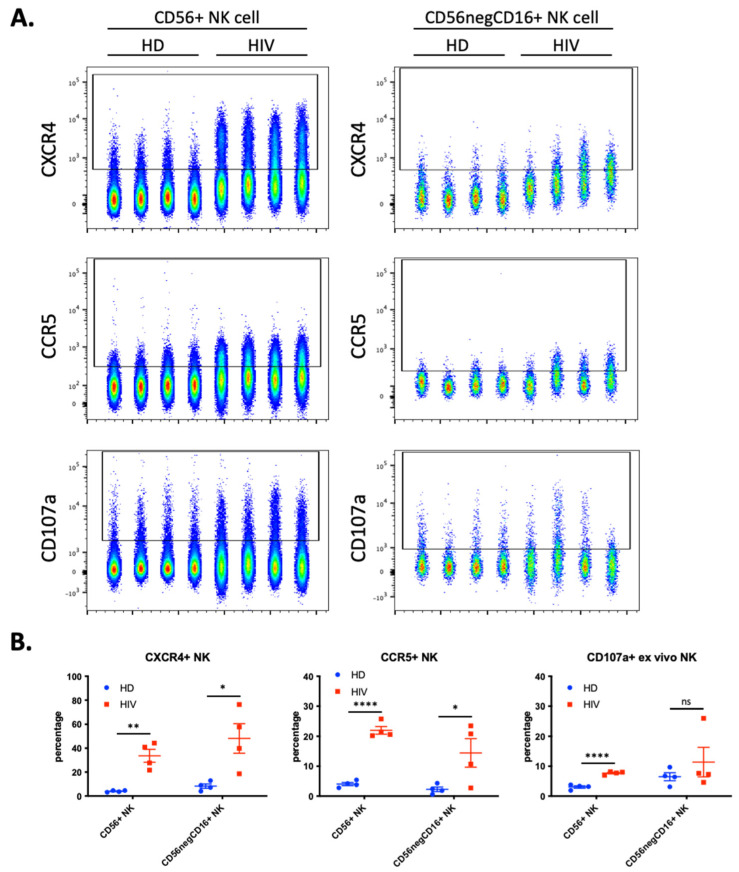
Enhanced trogocytosis and ex vivo degranulation activity on NK cells derived from four viremic, untreated, HIV-1 patients compared to four healthy donors (HD). (**A**) Level of CXCR4 (top), CCR5 (middle) and ex vivo degranulation (bottom panels) presented in parallel on two populations of NK cells: CD56^+^ NK (40,000 cells per sample; left panels) and CD56negCD16^+^ (2500 cells per sample; right panel). (**B**) Quantitative dot plots displaying the percentage of CXCR4^+^ (left), CCR5^+^ (middle) and CD107a^+^ NK (right) cells. Statistical significance was determined by un-paired *t*-test between HD vs. HIV-1; * *p* ≤ 0.05, ** *p* ≤ 0.01, and **** *p* ≤ 0.0001.

**Figure 6 vaccines-10-00688-f006:**
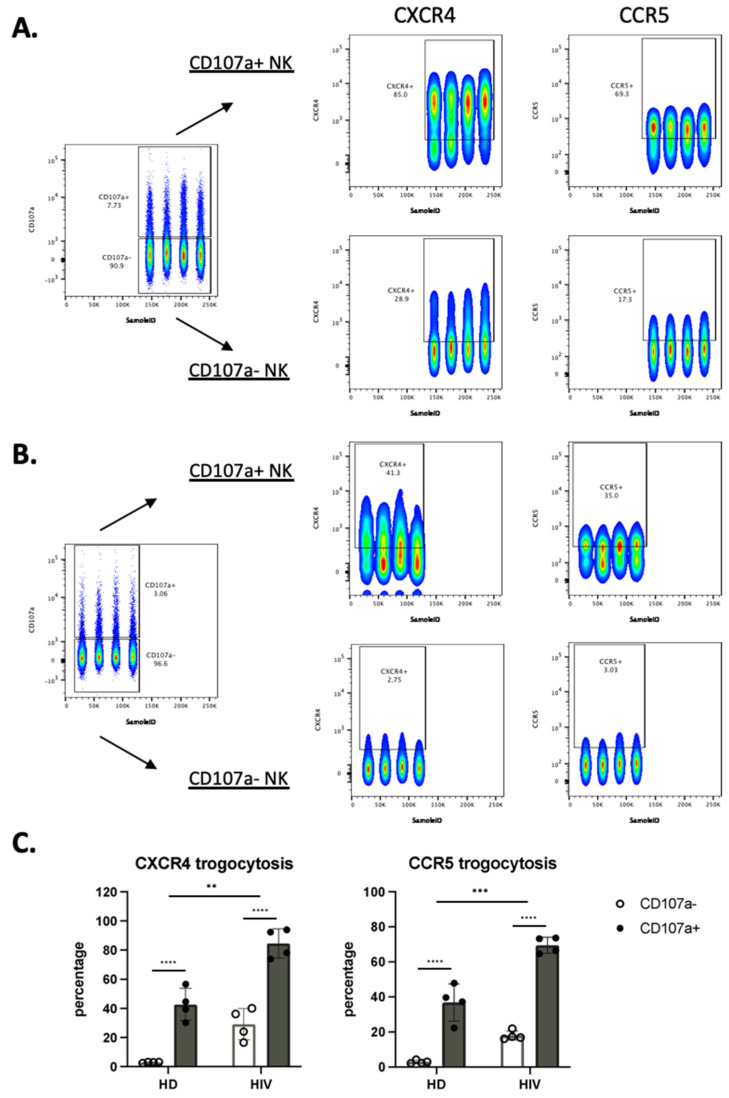
NK cells that have degranulated in viremic HIV-1^+^ patients performed enhanced trogocytosis. (**A**,**B**) Comparison of trogocytosis on CD107a^+^ vs. CD107a^−^ on CD56^+^ NK cells in viremic HIV-1 patients (**A**) and HD (**B**). (**C**) Percentage of CXCR4 and CCR5 expression on CD56^+^ NK cells expressing or not CD107a. (**D**,**E**). CD56^−^CD16^+^ NK cells in viremic HIV-1 patients performed enhanced trogocytosis. Comparison of trogocytosis on CD107a^+^ vs. CD107a- on CD56^−^CD16^+^ NK cells in HIV-1^+^ patients (**D**) and HD (**E**). (**F**) Percentage of CXCR4 and CCR5 expression on CD56^−^CD16^+^ NK cells expressing or not CD107a. Statistical significance was carried by two-way ANOVA between HD vs. HIV-1 and CD107a^−^ vs. CD107a^+^; * *p* ≤ 0.05, ** *p* ≤ 0.01, *** *p* ≤ 0.001 and **** *p* ≤ 0.0001.

**Figure 7 vaccines-10-00688-f007:**
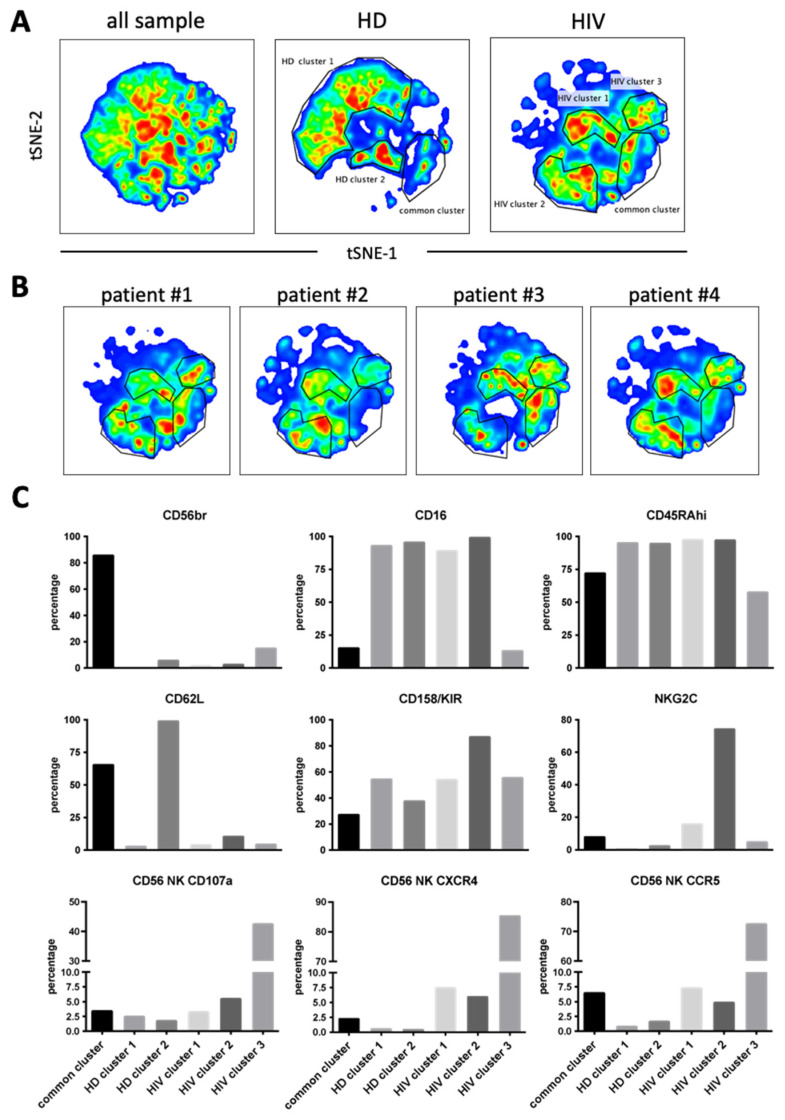
Cluster analysis on CD56^+^ NK cells by tSNE mapping in viremic patients. The tSNE map was built from 40,000 pre-gated CD56^+^ NK per sample (HD: *n* = 4, HIV-1: *n* = 4) based on 12 surface markers (Appendix A). (**A**) General tSNE map from all samples (left), HD samples (middle) and HIV-1^+^ samples (right) with identification of group-specific clusters. (**B**) tSNE replica showing clusters of each viremic HIV-1 patient. (**C**) Phenotypical characters of each cluster as identified in (**A**). Bar graphs represent the proportion of positive cells for each of the surface markers from the whole of events analyzed in each cluster.

**Figure 8 vaccines-10-00688-f008:**
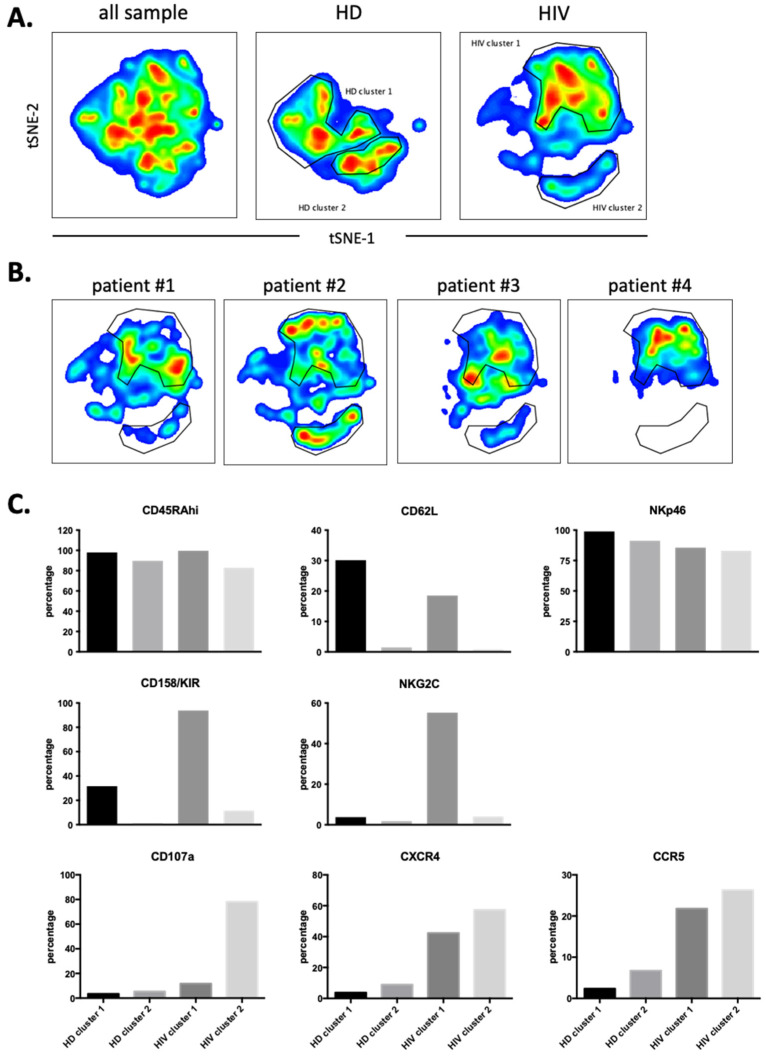
Cluster analysis on CD56negCD16^+^ NK cells by tSNE mapping in viremic patients. tSNE map was built from 2500 pre-gated CD56^neg^CD16^+^ NK per sample (HD: *n* = 4, HIV-1: *n* = 4) based on 12 surface markers (Appendix A). (**A**) General tSNE map from all samples (left), HD samples (middle) and HIV-1^+^ samples (right) with identification of group-specific clusters. (**B**) tSNE replica showing clusters of each HIV-1^+^ patients. (**C**) Phenotypical characters of each cluster as identified in (**A**). Bar graphs represent the proportion of positive cells for each of the surface markers from the whole of events analyzed in each cluster.

## Data Availability

Not applicable.

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
