# Peer review of "NK Cells Acquire CCR5 and CXCR4 by Trogocytosis in People Living with HIV-1"

_vaccines, 2022, doi:10.3390/vaccines10050688_

Round 1

Reviewer 1 Report

The title and abstract of this publication assume that the acquisition by NK cells of diverse markers better known on other cell types must be by trogocytosis. However there are relatively few papers in the literature to support this claim, and many studies based on mRNA suggest local production.

The first section of the Results claims uptake of a fluorescent dye, CD4 and CCR5 by a proportion of NK cells from a single donor….about whom we are told nothing. The data are in Supp Fig 1 and the presentation is cryptic.

Supp Fig 2 purports to show uptake of CXCR4 by NK cells from MT4 cells. However the NK cells are defined only by FS vs SSc. The proportions of CXCR4 positive cells are illegible but the patters look similar with and without MT4 cells. The data obtained with ex vivo NK cells is more convincing but it isn’t clear why CD56+ cells were purified and not simply gated. The latter would allow verification that the cells are CD3 negative…ie: not NK T-cells. The data are also only derived from a single donor and there is no control to establish whether CXCR4 expression on the stimulating cells (MT4) is critical. Expression of CD4 on the NK cells remained poor.

The next section “demonstrates” that CXCR4 is taken up by trogocytosis using an expression construct carrying CCR5. The leap across chemokines is not addressed. There is also no consideration of the possibility that the construct may exist in the culture and be taken up directly by NK cells.

The manuscript then uses UMAPs to distinguish NK cells from HIV px and healthy  donors. We are told that the maps are based on “12 NK markers”. These are shown in Supp Fig 3 and listed in the M&M…however they include CD4, CD45RA, CD45RO, CD62L and CCR5 – these are not “NK markers” so how can we be sure that the gates are exclusively NK cells.

HIV clusters 1 and 2 are defined in Fig 4B….leaving me to wonder what purpose was served by the UMAP analysis. Fig 4B & C do not specify the characteristics expression of the mother/source population.

Fig 4C shows clear gating of NKG2C+ cells in HIV patients (see below re CMV as a possible confounder). In Fig 4C, the CCR5 gating is much less distinct.

Fig 5 assumes that expression of CDCR5 and CXCR4 implies trogocytosis. The mother/source populations are again not defined. These HIV px are described as viraemic….does that mean “untreated”?

The work here includes CD56-CD16+ NK cells so it is difficult to merge the findings with the earlier experiments based on NK cells purified solely on the basis of CD56 expression.

We then return to the cluster analyses…leaving me unclear why we have returned and didn't consider this earlier. I am guessing that Figs 7C and 8C are about HIV px (it isn’t specified) …. And why no error bars (the tSNE-1 plots show big differences)?

Overall: I don't find the argument that this is trogocytosis convincing. Eg: it is not supported by evidence showing an absence of CCR5/CXCR4 mRNA

Finally:

  1. there is no consideration of the role of CMV. The literature contains many detailed reports of the induction of NKG2C by CMV? It is likely that the patients had a higher burden of CMV than the HD. This should be evaluated and discussed.
  2. Deletions in the genes encoding CCR5 and NKG2C exist in European populations. Heterozygosous carriage of these is common and reduced expression.

Author Response

The title and abstract of this publication assume that the acquisition by NK cells of diverse markers better known on other cell types must be by trogocytosis. However there are relatively few papers in the literature to support this claim, and many studies based on mRNA suggest local production.

The first section of the Results claims uptake of a fluorescent dye, CD4 and CCR5 by a proportion of NK cells from a single donor….about whom we are told nothing. The data are in Supp Fig 1 and the presentation is cryptic.

As described in the text, the aim of this supplemental figure was to show that NK cells can perform trogocytosis on T cell lines.  We have now included in the text that the donor is a healthy donor. In addition, we have also included in the supplemental figure 1 legend that NK cells are derived from a healthy donor.

Supp Fig 2 purports to show uptake of CXCR4 by NK cells from MT4 cells. However the NK cells are defined only by FS vs SSc. The proportions of CXCR4 positive cells are illegible but the patters look similar with and without MT4 cells. The data obtained with ex vivo NK cells is more convincing but it isn’t clear why CD56+ cells were purified and not simply gated. The latter would allow verification that the cells are CD3 negative…ie: not NK T-cells. The data are also only derived from a single donor and there is no control to establish whether CXCR4 expression on the stimulating cells (MT4) is critical. Expression of CD4 on the NK cells remained poor.

Supplemental Fig. 2 shows that is relatively easy to distinguish primary NK cells and the MT4 cell line.

The interest of purifying NK cells before coculturing them is to discard the hypothesis of a role of another PBMC subpopulation in trogocytosis.

Yes, CD4 transfer is very poor, probably this antigen is not easily trogocytosed. This could explain why we do not observe CD4 trogocytosis ex vivo.

We now provide a new Supp Fig 2 where the proportions of CXCR4-positive cells among CD56low cells are clearly indicated. The patterns of CXCR4 expression look similar with and without MT4 cells, but the proportions of CD56low cells positive for CXCR4 are clearly increased (11,9% to 45.3%). CXCR4 densities on CD56low cells also increased from a mean fluorescence intensity of 94 to a mean fluorescence intensity of 190 arbitrary units.

The next section “demonstrates” that CXCR4 is taken up by trogocytosis using an expression construct carrying CCR5. The leap across chemokines is not addressed. There is also no consideration of the possibility that the construct may exist in the culture and be taken up directly by NK cells.

There is no possibility that the construct may exist in the culture and be taken up directly by NK cells, since MT4 cells were extensively washed after transduction, then amplified with several rounds of cell culture with fresh medium, frozen, thawed, washed, and cultured again. Moreover, our experience, and that of all our collaborators, is that primary NK cells are very difficult to transfect. Hence, it is also extremely difficult that just by adding the plasmid into the culture media will induce its expression in NK cells

We use CCR5-GFP as a test to investigate if NK cells can trogocytose antigens from T cells in vitro. We assume that if CCR5 was gained, CXCR4 can also be obtained.

The manuscript then uses UMAPs to distinguish NK cells from HIV px and healthy  donors. We are told that the maps are based on “12 NK markers”. These are shown in Supp Fig 3 and listed in the M&M…however they include CD4, CD45RA, CD45RO, CD62L and CCR5 – these are not “NK markers” so how can we be sure that the gates are exclusively NK cells.

As we described in the text, we investigate these markers for the following reasons:

CD4 to investigate if this Ag is trogocytosed

CD45RA and CD45RO as markers of the antitumor NK cells. The idea was to investigate if these markers are also present in anti-HIV cells

CD62L, in fact, this is a homing development BJK cell marker (it is not only expressed in T cells)

CCR5 to investigate if NK cells have performed trogocytosed in this Ag.

In summary, the idea was to cluster NK cells in clusters than COULD express those markers. Specific NK cell markers are very few, and it was not our goal to investigate how they cluster.

As stated in the Figure 3 legend, we analyzed: from 40000 pre-gated NK cells (CD14-CD19-CD3-CD56+ phenotype)

As stated in Fig. 7; we analyzed: “40000 pr-gated CD56+ NK per sample

As stated in Fig. 8 legend, we analyzed “2500 pre-gated CD56negCD16+ NK per sample”

Hence, we analyzed CD56+ cells. This is the most extensive marker use for NK cell identification. Moreover, the pattern of expression of other markers confirm that these cells express the typical NK cell markers (see expression pattern through the MS). Obviously in Fig. 8 we did not use CD56, but CD16 to identify lymphocytes that could be NK cells. However, here we are obviously more cautious about the origin of these cells.

HIV clusters 1 and 2 are defined in Fig 4B….leaving me to wonder what purpose was served by the UMAP analysis. Fig 4B & C do not specify the characteristics expression of the mother/source population.

Clusters are defined in Fig.3B and analyzed in Figs 4A & B for the antigen expression pattern. To avoid confusion in this figure we add the following sentence:

“In view of the relationship between a viral infection, i.e. CMV, and NKG2C expression [20,21], we directly analyzed NKG2C expression in our samples,”

Regarding the populations that have been analyzed, we have modified the figure 4 legend to clarify this point:

Fig. 4B legend states now: “Manual gating on CD57 and NKG2C for all concatenated NK cells from the HIV patients, which were divided into 4 clusters as described in Figure 3B.

Fig 4C shows clear gating of NKG2C+ cells in HIV patients (see below re CMV as a possible confounder). In Fig 4C, the CCR5 gating is much less distinct.

Fig 5 assumes that expression of CDCR5 and CXCR4 implies trogocytosis. The mother/source populations are again not defined. These HIV px are described as viraemic….does that mean “untreated”?

Viraemic patients were indeed treatment naïve. We have included the sentence “4 viremic, untreated, HIV patients” in the Fig. 5 legend to clarify this point.”

The work here includes CD56-CD16+ NK cells so it is difficult to merge the findings with the earlier experiments based on NK cells purified solely on the basis of CD56 expression.

We had already described in the MS that:

“During chronic HIV-1 infection, there is an expansion of a CD56- NK cell population that can constitute up to half of the peripheral NK cells and are functionally impaired [22–24]. This NK cell subset is also found in healthy individuals [25,26], but it is unknown to what extent this population represents a similar or distinct phenotype in patients and in healthy individuals. Hence, we analyzed the expression of the abovementioned markers in the CD56-CD16+ population. “

We fully agree with the reviewer that it is complicated to fully identify this population as bona-fide NK cells. However, in view of the large description of this population in the previous literature (see our previous paragraph), we consider that it is interesting for the field to analyze this population.

We then return to the cluster analyses…leaving me unclear why we have returned and didn't consider this earlier. I am guessing that Figs 7C and 8C are about HIV px (it isn’t specified) …. And why no error bars (the tSNE-1 plots show big differences)?

We have now included in both figure legends that these patients are viremic.

Regarding Figs 7C- 8C, the bar plots from these figures represent the proportion of positive cells for each of the surface markers from the whole of events analyzed in each cluster. This means there is one measurement for each cluster, therefore standard error is not available for this approach of analysis (the point is to compare inter-cluster variation, but not aiming to gain a statistical information).

The figure legends 7 and 8 have been modified accordingly to clarify this point.

Overall: I don't find the argument that this is trogocytosis convincing. Eg: it is not supported by evidence showing an absence of CCR5/CXCR4 mRNA

The process of trogocytosis is well documented in several types of lymphocytes. In vitro we show here that NK cells can trogocytosed T cell antigen, e.g. the appearance of CCR5/EGFP at the surface of the purified NK cells cocultured with CCR5/EGFP-MT4 cells, but not at the surface of NK cells cocultured with LacZ-MT4 cells. In vivo we show an extremely clear relationship between antigen expression, i.e. CCR5 and CXCR4 and degranulation. In the absence of any indication that NK cell degranulation leads to the expression of these markers, we support that trogocytosis is the physiological phenomenon underlining our observations.

Finally:

  1. there is no consideration of the role of CMV. The literature contains many detailed reports of the induction of NKG2C by CMV? It is likely that the patients had a higher burden of CMV than the HD. This should be evaluated and discussed.

The reviewer is right. HIV patients harbor more often CMV than HIV-negative persons. Yet, CMV does not induce CXCR4 expression (J Virol. 2018 Feb 12;92(5):e01981-17). Hence, we have included the following sentence in the text:

“Of note, although HIV patients harbor more often CMV than HIV-negative persons, CMV does not induce CXCR4 expression [22].”

In fact, we have previously answered a similar question and added the sentence:

“In view of the relationship between a viral infection, i.e. CMV, and NKG2C expression [20,21], we directly analyzed NKG2C expression in our samples,”

Moreover, in the original submission we had largely discussed the role of NKG2C expression in our results. For example, this paragraph in the discussion:

“In agreement, the cells present in HIV cluster 2 (described here in Fig. 4) exhibit NKG2C and KIRs and have performed trogocytosis on CCR5. This resembles to the recently described memory-like NK cell population in HIV patients or macaques model [20,21,38,40–42], which shows increased effector functions when reencountering viral antigens. This would explain why they carry T cell markers, i.e. they obtained them by trogocytosis after encountering HIV-infected T cells. However, we observed the cells that have trogocytosed lacked CD57 expression. It is possible that when evolving from memory (CD57+) to effector cells, CD57 is lost. Interestingly, in active viremic patients, we found a negative relationship between degranulation and trogocytosis versus NKG2C expression. Probably, these patients have not generated yet NKG2C+ memory-like NK cells or in active viremia, this population does not have time to established itself. Interestingly, in viremic patients the cluster that has degranulated and performed trogocytosis are CD56dim cells with low CD16 expression. This decrease in CD16 is found in NK cells after encountering with target cells [43]. They expressed also low CD45RA levels, which could be due to increase expression of CD45RO. CD45RO is linked to antitumor NK cell function in hematological cancer patients [4,8–10]. In summary, in HIV patients the populations of memory and effector NK cells are phenotypically different. Antiviral memory NK cells express NKG2C whereas antiviral effector NK cells lack NKG2C, but can be recognized by degranulation and by their capacity to perform trogocytosis. »

2. Deletions in the genes encoding CCR5 and NKG2C exist in European populations. Heterozygosous carriage of these is common and reduced expression.

The level of CCR5 and CXCR4 expression we reported in HD is consistent with the levels reported in the literature, and much lower than the levels we observed in people living with HIV. Moreover, we observed CCR5 and CXR4 overexpression on NK cells in all the viremic patients we analyzed. This makes very unlikely that the differences we observed are genetically-driven.

Reviewer 2 Report

The manuscript by Dang-Nghiem et al reports the uptake of CCR5 and CXCR4 receptors by NK cells. The authors prove via experimentation this uptake of the CCR5 and CXCR4 receptors is via trogocytosis. The authors also highlight a probable link between the degranulation and trogocytosis. The study is well executed and is straightforward and the discussion of the results is thorough and informative. The authors have strategically employed a number of experiments to support their aims. One minor comment- page 4 line 157: The authors mention that 59% of the cocultured CD56-positive lymphoid cells became CXCR4-positive, should this be 62.5% instead?

Author Response

The manuscript by Dang-Nghiem et al reports the uptake of CCR5 and CXCR4 receptors by NK cells. The authors prove via experimentation this uptake of the CCR5 and CXCR4 receptors is via trogocytosis. The authors also highlight a probable link between the degranulation and trogocytosis. The study is well executed and is straightforward and the discussion of the results is thorough and informative. The authors have strategically employed a number of experiments to support their aims. One minor comment- page 4 line 157: The authors mention that 59% of the cocultured CD56-positive lymphoid cells became CXCR4-positive, should this be 62.5% instead?

We thank the reviewer for these kind comments and we have modified the text accordingly.

Reviewer 3 Report

The manuscript “NK cells acquire CCR5 and CXCR4 by trogocytosis in people living with HIV-1” by Dang Nghiem Vo, et al, describes specific NK cell subsets that have acquired CCR5 and CXCR4 markers by trogocytosis in HIV-1 positive patients. The manuscript is well written and easy to follow. The data presented, for the most part, supports their conclusions. A few comments that need to be addressed.

  • HIV-1 infected patient population: Please clarify if the patients categorized as nontreated and viremic were treatment naïve patients.
  • Data showing the increased expression of CCR5 and CXCR4 and trogocytosis by NK cells (Fig. 5 and Fig.6) and cluster analysis (Fig. 7 and Fig.8) is provided only for viremic HIV-1 patients. No supporting data is provided for aviremic HIV-1 patients under stable antiretroviral therapy. The authors should provide additional information to show that the difference between the viremic and aviremic HIV-1 patients.
  • English Language usage: Please carefully check the manuscript once more and correct errors. e.g. Please change line 346 (Probably, these patients have not generated yet NKG2C+ memory-like NK cells) to “Probably, these patients have not yet generated NKG2C+ memory-like NK cells”.
  • The authors use HIV-1 and HIV interchangeably. Since the authors used only samples from HIV-1 mono-infected patients and HIV-1 for the in vitro studies, the term HIV should be changed to HIV-1 throughout the manuscript.

Author Response

The manuscript “NK cells acquire CCR5 and CXCR4 by trogocytosis in people living with HIV-1” by Dang Nghiem Vo, et al, describes specific NK cell subsets that have acquired CCR5 and CXCR4 markers by trogocytosis in HIV-1 positive patients. The manuscript is well written and easy to follow. The data presented, for the most part, supports their conclusions. A few comments that need to be addressed.

  • HIV-1 infected patient population: Please clarify if the patients categorized as nontreated and viremic were treatment naïve patients.

Viremic patients were indeed treatment naïve. We have included the sentence “4 viremic, untreated, HIV patients” in the Fig. 5 legend to clarify this point.”

  • Data showing the increased expression of CCR5 and CXCR4 and trogocytosis by NK cells (Fig. 5 and Fig.6) and cluster analysis (Fig. 7 and Fig.8) is provided only for viremic HIV-1 patients. No supporting data is provided for aviremic HIV-1 patients under stable antiretroviral therapy. The authors should provide additional information to show that the difference between the viremic and aviremic HIV-1 patients.

In fact, Fig. 4C shows CCR5 expression in 20% of patient’s NK cells, compared to 7% of HD. Fig. 4A shows that HIV cluster 2 contains more of these cells. The proportion of CCR5+ NK cells is higher in viremic patients, but unfortunately both cohorts were analyzed at different times and using other markers. Hence, we cannot formally compare all 4 populations, i.e. HD for cohort 1, HD for cohort 1, viremic patients and aviremic patients.

  • English Language usage: Please carefully check the manuscript once more and correct errors. e.g. Please change line 346 (Probably, these patients have not generated yet NKG2C+ memory-like NK cells) to “Probably, these patients have not yet generated NKG2C+ memory-like NK cells”.

We have corrected that and other English errors.

  • The authors use HIV-1 and HIV interchangeably. Since the authors used only samples from HIV-1 mono-infected patients and HIV-1 for the in vitro studies, the term HIV should be changed to HIV-1 throughout the manuscript.

We have followed the reviewer’s recommendation.

Reviewer 4 Report

In the paper NK cells acquire CCR5 and CXCR4 by trogocytosis in people living with HIV-1,  the authors present robust data indicating that NK, in HIV donors, presents a higher surface expression of CCR5 and CXCR4. The possible mechanism is trogocytosis, already described for NK cells.

The paper is very clear and easy to read. In general, the figures are clear and support the claimed data. Overall the paper is very well presented.

I have in any case few questions/considerations:

---You used UMAP for some analysis and for others t-SNE. The two methods are really not the same, the mathematics behind them are very different also if,  in theory, they should produce the same results. In particular, I don t see any advantage in using t-SNE anymore. The math behind UMAP is far more robust and  UMAP results are produced way faster than t-SNE. I would like you to point out the reasons behind the use of t-SNE for some specific analysis instead of UMAP.

---In vitro, NKs were able by trogocytosis, to acquire CCR5 and CXCR4 when cultured with the cell line MT4 and CEM (eventually transfected with HIV vector CCR5/EGFP). Ex vivo, the NKs from viremic donors present a higher frequency of cells positive for CCR5 and CXCR4.

It would be formally more elegant to repeat the same experiment using as feeder cells PBMNC (NK depleted) (or CD4) from HIV-infected but ART-treated individuals.

For example, NK and PBMNC (NK depleted) (or CD4) from a treated individual could be isolated and the PBMNC (NK depleted) (or CD4) could be stimulated for a few days in order to reactivate the virus and spread the infection (one-week CD3/CD28/IL2 stimulation should be sufficient, or even with PMA/ion). This stimulated PBMNC (NK depleted) (or CD4) could be cocultured with resting NK. If those acquire CCR5 and CXCR4 it would be definitive proof of trogocytosis.

---CCR5 and CXCR4, among the rest, are important for the homing. In particular, on CD4 T cells, CCR5 and CXCR4 are responsible for the homing to the Lymph Nodes. Do you had/have the possibility to check if in LN of infected individuals the NK/CCR5+/CXCR4+ are enriched compared to HD?

---3 specific NK populations were detected in this study in HIV-infected individuals (fig 7) with specific signatures about the expression of CCR5, CXCR4, CD56, CD62L, NKG2C etcetc. It is strange that you didn’t look at other very important surface molecules, in particular the inhibitors ones, like PD1/PDL1, CTLA4, BTLA, CD160, IDO, KLRG1, LAG-3, LILRB1, TRAIL, TIM-3, and targets. After all, the NKs have the potency to limit the HIV infection BUT in the end, they fail this in vivo. The trogocytosis of such checkpoint molecules could explain this phenomenon.

---I would like also you to discuss why the process of trogocytosis leads to the transfer of CCR5 and CXCR4 on the NK but not to the transfer of CD4. It looks to be an important matter that deserves some further study/comment.

---Fig2 supplement is very poor.

Author Response

In the paper NK cells acquire CCR5 and CXCR4 by trogocytosis in people living with HIV-1,  the authors present robust data indicating that NK, in HIV donors, presents a higher surface expression of CCR5 and CXCR4. The possible mechanism is trogocytosis, already described for NK cells.

The paper is very clear and easy to read. In general, the figures are clear and support the claimed data. Overall the paper is very well presented.

I have in any case few questions/considerations:

---You used UMAP for some analysis and for others t-SNE. The two methods are really not the same, the mathematics behind them are very different also if,  in theory, they should produce the same results. In particular, I don t see any advantage in using t-SNE anymore. The math behind UMAP is far more robust and  UMAP results are produced way faster than t-SNE. I would like you to point out the reasons behind the use of t-SNE for some specific analysis instead of UMAP.

These results of high-dimensional analysis based on flow cytometry data were generated by t-SNE/UMAP plugins implemented in FlowJo software (please see Material&Methods 2.6 section for detail description of the workflow). Throughout the manuscript, the standard UMAP dimensional reduction analysis was performed for multiparameter FACS data, except results presented in Fig 7-8 where t-SNE was performed in early 2018 when the UMAP algorithm has not yet been available (McInnes, Healy and Melville, 2018). We agreed with the reviewer’s comment that UMAP was demonstrated to be superior to t-SNE when preservation of global structure is considered to be critical. However, here in NK cell phenotypic analysis presented in Fig 7-8, such global structure of multiple clustering pattern did not exist. Therefore we believed that the difference between UMAP versus t-SNE in the final output is minimal, if any.

---In vitro, NKs were able by trogocytosis, to acquire CCR5 and CXCR4 when cultured with the cell line MT4 and CEM (eventually transfected with HIV vector CCR5/EGFP). Ex vivo, the NKs from viremic donors present a higher frequency of cells positive for CCR5 and CXCR4.

It would be formally more elegant to repeat the same experiment using as feeder cells PBMNC (NK depleted) (or CD4) from HIV-infected but ART-treated individuals.

For example, NK and PBMNC (NK depleted) (or CD4) from a treated individual could be isolated and the PBMNC (NK depleted) (or CD4) could be stimulated for a few days in order to reactivate the virus and spread the infection (one-week CD3/CD28/IL2 stimulation should be sufficient, or even with PMA/ion). This stimulated PBMNC (NK depleted) (or CD4) could be cocultured with resting NK. If those acquire CCR5 and CXCR4 it would be definitive proof of trogocytosis.

This is an interesting remark. Maybe this might be explained by the fact that our in vitro coculture was brief, whereas in vivo NK cells are exposed for a longer time to CCR5+ and/or CXCR4+ infected cells. Moreover, it may be argued that in vivo circulating NK cells are exposed to more CCR5+ and/or CXCR4+ circulating infected cells than in vitro.

We thank the reviewer for proposing those experiments. We have already shown that NK cells can trogocytose CCR5 and CXCR4 in vitro by using the CCR5/EGFP construct that we introduced in MT4 cells. This excludes the endogenous expression of CCR5 by NK cells. We believe that the experiments proposed by the reviewer, although very elegant, would not add a stronger argument, since in this assay it could be argued that CCR5 acquisition on NK cells is due to endogenous production.

---CCR5 and CXCR4, among the rest, are important for the homing. In particular, on CD4 T cells, CCR5 and CXCR4 are responsible for the homing to the Lymph Nodes. Do you had/have the possibility to check if in LN of infected individuals the NK/CCR5+/CXCR4+ are enriched compared to HD?

Unfortunately, we have not had the opportunity to have such biopsies, but we will try to obtain them for future analysis.

---3 specific NK populations were detected in this study in HIV-infected individuals (fig 7) with specific signatures about the expression of CCR5, CXCR4, CD56, CD62L, NKG2C etcetc. It is strange that you didn’t look at other very important surface molecules, in particular the inhibitors ones, like PD1/PDL1, CTLA4, BTLA, CD160, IDO, KLRG1, LAG-3, LILRB1, TRAIL, TIM-3, and targets. After all, the NKs have the potency to limit the HIV infection BUT in the end, they fail this in vivo. The trogocytosis of such checkpoint molecules could explain this phenomenon.

We are limited for the number of antigens that we can analyze in our cytometer. We need to target several antigens to exclude other populations, i.e. CD19 (B cells); CD14/CD33 (myeloid cells), CD3 (T cells). In summary, although we have a panel with 20 colors, finally we only can asses several Ags. However, we agree that a larger study could be very interesting to develop in a subsequent cohort of patients.

---I would like also you to discuss why the process of trogocytosis leads to the transfer of CCR5 and CXCR4 on the NK but not to the transfer of CD4. It looks to be an important matter that deserves some further study/comment.

We will also really like to know this information. The main problem is that we do not know the mechanism of trogocytosis and, hence, we do not understand why certain plasma membrane proteins are easily transfer or presented and others are not. We have included the following sentence in the Discussion section: “Why are certain plasma membrane molecules trogocytosed and not others? This is unknown, and it is probably related to the unknowledge of the mechanism(s) of trogocytosis [5,28].”

Unfortunately, we can not give additional information on this subject.

---Fig2 supplement is very poor.

We have included a new supplemental Fig. 2.

Round 2

Reviewer 1 Report

My previous comments have been argued with some success but this has had little effect on the ms. I would like to see a section in the Discussion that deals with "limitations to this study". It should properly evaluate the possible importance of genetics (eg the CCR5 deletion), extrapolations between CCR5 and CXCR4 (cf Fig 5), and CMV - plus the absence of data addressing mRNA for the chemokine receptors. The "Conclusions" should be written with more care noting that the NK cells gain CCR5 & CXCR4 but evaluating the evidence that this is by trogocytosis.

Author Response

As requested by the reviewer, we have included the following paragraph in the discussion:

Trogocytosis is becoming a field of intense study to understand the communication/interaction of tumor and immune cells, which can affect the clinical outcome as recently shown by trogocytosis of PD-1 by NK cells in cancer patients [45]. Being sure that ex vivo observations properly mirror the in vivo physiology is extremely difficult for all this kind of studies, and hence, our study has some limitations. Besides trogocytosis, CCR5 and CXCR4 overexpression on NK cells we report here in nontreated HIV patients could have other causes. A genetic cause is unlikely. D32CCR5 heterozygosity is a known cause of low cell surface CCR5 density. Yet, the frequency of D32CCR5/WTCCR5 individuals in France is 0.18 [46], so the probability for the 4 HD of each cohort that we have analyzed to be D32CCR5 heterozygous is extremely low. Coinfections could also play a role. For instance, human CMV infection, which is more frequent in HIV-positive than in HIV-negative individuals, is associated with an increased frequency of NKG2C+ NK cells [47]. Yet, as mentioned, CMV does not induce CXCR4 expression. It would be interesting to support our proposal that CCR5 and CXCR4 overexpression on NK cells in HIV patients is due to trogocytosis by quantifying CCR5 and CXCR4 mRNA in NK cells to discard the hypothesis of an endogenous production. However, this will imply the sorting of a small number of cells and the corresponding ex vivo manipulation that could affect gene expression.

We have changed the Conclusion to follow reviewers’ comments:

NK cells play a major role in the antiviral immune response, including against HIV-1. We identified specific NK cell subsets that expressed CCR5 and CXCR4, but barely CD4, T cell antigen markers on their plasma membrane. We show a strong association of degranulation and trogocytosis ex vivo suggesting that NK cells are eliminating or trying to eliminate HIV-1 infected T cells. By UMAP (Uniform Manifold Approximation and Projection), we show that aviremic HIV-1 patients have unique NK cell clusters that encompass for cells expressing CCR5, NKG2C and KIRs, but lack CD57 expression. Viremic patients have a larger proportion of CXCR4+ and CCR5+ NK cells than healthy donors (HD) and this subset lacks NKG2C expression. Therefore, our results strongly suggest that NK cells can gain CCR5 and CXCR4 by trogocytosis in vivo, which depends on degranulation and does not always correlate with NKG2C expression.